# A Geospatial Modelling Approach to Understand the Spatio-Temporal Impacts of Grazing on Soil Susceptibility to Erosion

**Fabiellen C. Pereira** [1,2,*] , **Mitchell Donovan** [3] , **Carol M. S. Smith** [2,4] , **Stuart Charters** [2,5] , **Thomas M. R. Maxwell** [1,2] and **Pablo Gregorini** [1,2]

1    Department of Agricultural Science, Faculty of Agriculture and Life Sciences, Lincoln University, Lincoln 7674, New Zealand
2    Centre of Excellence Designing Future Productive Landscapes, Lincoln University, Lincoln 7674, New Zealand
3    AgResearch Limited Invermay Agricultural Centre Puddle Alley, Mosgiel 9053, New Zealand
4    Department of Soil & Physical Sciences, Faculty of Agriculture and Life Sciences, Lincoln University, Lincoln 7674, New Zealand
5    School of Landscape Architecture, Faculty of Environment, Society and Design, Lincoln University, Lincoln 7674, New Zealand
*    Correspondence: fabiellen.pereira@lincolnuni.ac.nz

**Abstract:** Grazing management to reduce soil erosion is paramount for preserving and enhancing grassland health under pastoral livestock production systems. However, as the focus of these production systems is to increase productivity, the inclusion of the soil and its complexity in grazing management has been usually neglected. Detailed consideration of the soil spatio-temporal susceptibility to erosion may be best approached with simulation modelling. To understand and explore the spatio-temporal impact of grazing strategies on soil surface erosion, this work used a geospatial model approach in a high-country pastoral livestock production system in New Zealand as a case study. We modelled 45 scenarios characterized by different stock densities and occupation periods applied for each season of the year and for different livestock types: sheep, beef, and deer, producing a total of 540 scenarios. In addition, we included scenarios to represent ungrazed pastures for each season and the current grazing management of the case study station as the baseline for comparisons (resulting in a total of 545 scenarios). Spatio-temporal variation of natural soil superficial erosion from ungrazed pastures appears to be more relevant than the impact of manipulating grazing intensity and indicates that paddocks in our study area have different capacities to support grazing which also changes during seasons. Increases in occupation period seem more detrimental to soil erosion compared to increases in stock density, and cattle are the most detrimental stock type compared to sheep and deer. Our results suggest that grassland health can be enhanced in LUMGS by applying context-adjusted grazing management strategies according to the station spatio-temporal heterogeneity and susceptibility to erosion.

**Keywords:** grassland; grazing management; health; soil; pastoral livestock production systems

## 1. Introduction

Soil is essential for the performance of pastoral livestock production systems, not only for providing nutrients and water for plant biomass production, but also for providing ecosystem services that support grasslands and human life. Soil is one of the largest sinks of C storage/sequestration, stores and filters water, controls pests and diseases and supports a biodiverse habitat for organisms [1,2]. The ability of soil to fulfil its functions is determined by its quality [3]. Soil quality is threatened by human activities that affect soil physical, chemical, and biological properties, causing soil degradation and loss of productivity [4,5]. Dominant causes of soil degradation are attributed to grazing animals [6], due to the pressure caused during treading and other grazing activities [7,8]. While grazing

animals also contribute to the ecological functions of the soil by providing natural fertiliser and promoting cover and protection through plant renewal [9,10], grazing pressure and frequency should be managed to avoid soil degradation. In doing so, we can maintain soil quality and productivity for the long-term provision of ecosystem services [1].

Soil deformation may occur when the physical force that animals exert on soils during treading is greater than the soil resistance capability [5]. The capacity of soil to support grazing depends on intrinsic soil properties and topographic factors, such as soil structure, moisture, texture, and slope angle. Soils on steep slopes are in general more sensitive to treading damage than those on flat surfaces, due to a finer texture and lower aggregate stability and organic matter content [11]. Those soil characteristics reduce the ability of the soil to withstand treading without critical loss of groundcover [12]. Soil texture and organic matter content influence the volume of soil moisture, which affects soil consistency and determines its susceptibility to deform or break [13]. Accordingly, soils that are typically wet in winter and spring such as Pallic soils are more prone to treading damage [7], due to a high soil moisture content [13]. The consequences of treading extend to changes in soil physical properties. Soil pore space, hydraulic conductivity, and infiltration rates are reduced, while bulk density increases, thus decreasing soil cohesion and strength and causing compaction (or pugging when soil is wet) and increasing its susceptibility to erosion [7,14]. As vegetation cover protects the soil against treading pressure [8,15] and rainfall interception [16], decreased ground cover caused by excessive defoliation and treading [6] is also a grazing impact that increases soil exposure and susceptibility to erosion in pastoral livestock production systems [14,17].

Grazing management should be adjusted to avoid exceeding the tread bearing capacity of soils. Heavy grazing pressure, which increases the force animals exert on soil, is more likely to reduce vegetation cover and physically damage soil [12]. In Zhang et al. (2019 [18]), increased grazing density during a period of occupation of six months significantly changed soil physical properties (bulk density, sand, clay, and silt content), mainly on steep lands. Similarly, in Pulido et al. (2018, [6]) higher stock density in continuous management affected pasture production and soil cover, leading to soil degradation. The species of animal also lead to different impacts. As soil compaction or pugging is a consequence of the force an animal imposes on the soil surface, which is dependent upon animal body weight, different species exert different forces, and therefore, affect soil properties and its susceptibility to degradation in different ways [5,7]. However, the effects of stock density and species also depend on the grazing strategies associated with the duration that the animals graze in an area. Pasture and soil degradation can be better controlled in a multi-paddock management as the grazing pressure is more homogeneous compared to a continuous management in which the unequal distribution of animals can lead to overgrazing [19]. The use of areas for a shorter period with a long period of recovery has been shown to improve soil quality and the resilience of grazing systems [9]. Still, the best strategy will depend on the capacity of the soil to withstand grazing, according to its properties.

Appropriate grazing management strategies to reduce soil damage and erosion during treading remains an open question. As soils respond to grazing animals distinctively, grazing management should be based on strategies that spatio-temporally suit a particular soil context. Modelling can support the design of the most appropriate grazing management for a particular area because it can simulate changes across an array of soils under various grazing strategies [14]. A model is a simplification of a reality that aims to identify circumstances that affect the possible outcomes of a system [20]. Thus, the use of modelling would be a quick and inexpensive approach to provide us insights about the response of soils to treading under different conditions and accelerate our understanding about grazing systems.

The revised universal soil loss equation (RUSLE) is the most widely applied model for estimating soil loss from surface erosion as it integrates important factors that contribute to soil erosion: soil erodibility (K), ground cover (C), rainfall erosivity (R), slope length (L), and steepness (S) [21–23]. However, the RUSLE model does not consider the grazing and treading factor. To improve the accuracy of the RUSLE model and understand the impact

of livestock on soil loss in New Zealand, a novel geospatial livestock treading and grazing model has been proposed by Donovan and Monaghan (2021 [14]). The model estimates soil loss by assessing changes in soil permeability and structural aggregates caused by compaction and pugging during treading. The model outputs are then included in a novel form of the RUSLE, which further incorporates seasonal ground cover, slope length steepness, and seasonal rainfall. Simulating grazing impact through modelling aids in the exploration of alternative grazing management strategies and the visualization of outcomes over different areas and climates whilst still considering the spatial heterogeneity of the landscape [9]. The Donovan and Monaghan (2021 [14]) model uses New Zealand national databases, considering the vast variety of soil, landcover, and topographic characteristics across the country and its outcomes are consistently accurate with measurements from field studies. Thus, we applied the treading and grazing model proposed by Donovan and Monaghan (2021 [14]) to assess changes in soil erosion resulting from the spatio-temporal impact of different stock densities, grazing occupation periods, and stock type in topographically distinct paddocks on a high-country pastoral livestock production system in New Zealand.

## 2. Materials and Methods

In this section, we start by describing the study area and the current grazing management (status quo) adopted there. Then, we describe the treading and grazing model proposed by Donovan and Monaghan (2021 [14]) in detail. Finally, we describe the grazing management strategies modelled in this work.

### 2.1. Lincoln University Mount Grand Station

The treading and grazing model was applied to the Lincoln University Mount Grand Station (LUMGS). The station is a high-country farm located close to Lake Hawea in Central Otago in the South Island (lat. 44°38′01.93″ S; long. 169°19′42.89″ E) of New Zealand (See Figure 1). The annual rainfall average is 703 mm, and the annual mean temperature is 10.6 °C (17.25 °C in summer and 2.25 °C in winter [24,25]. The farm encompasses 2131 ha of which 1602 ha are used as a pastoral system, while the remainder is a conservation area. Of the total area, 7% is flat, with the vast majority being undulating hill country with steep slopes reaching up to 65° over an elevation range of 350 m to 1400 m above the sea.

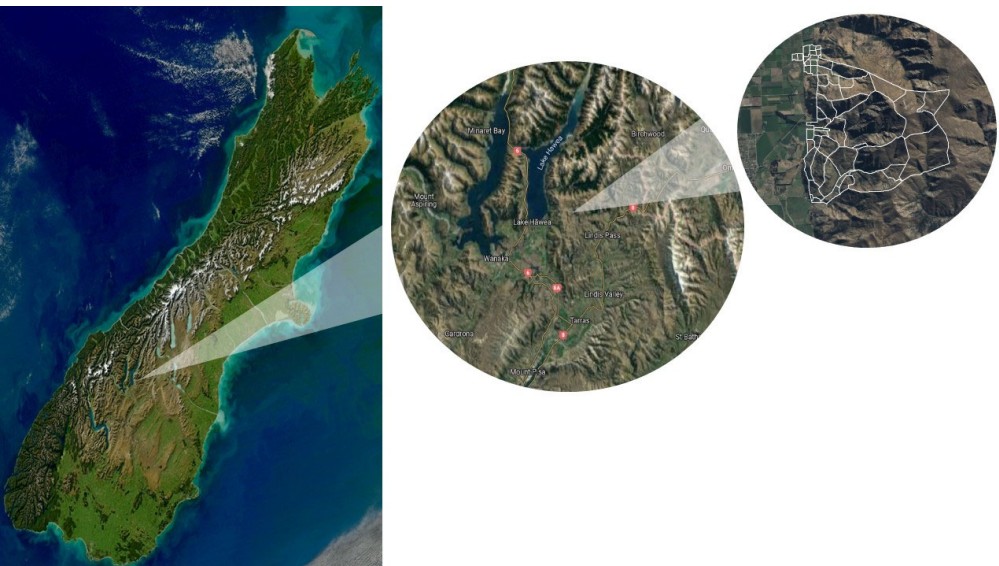

**Figure 1.** The geographic location of Lincoln University Mount Grand Station, latitude 44°38′01.93″ S; longitude 169°19′42.89″ E, and paddocks boundary. From left to right: The South Island of New Zealand (NZ), Lake Hawea—Central Otago, and the paddocks boundary of the station.

Soils of LUMGS are formed on Haast schist, loess, and alluvial gravels [26] and are predominantly classified as Brown soils (upland slopes developed on the schist bedrock) and Pallic soils (loess parent material), with a small proportion of rocky Raw soils occurring at higher elevation [27] (Figure 2). LUMGS soils are silt loam [28], shallow and stony with low water holding capacity [25]. On the upper and lower altitude hill paddocks 125–150 kg/ha of 30% Sulphur superphosphate fertilizer is applied every two years, while on the flat areas a topdressing 150–200 kg/ha of 30% Sulphur superphosphate fertilizer is applied every year.

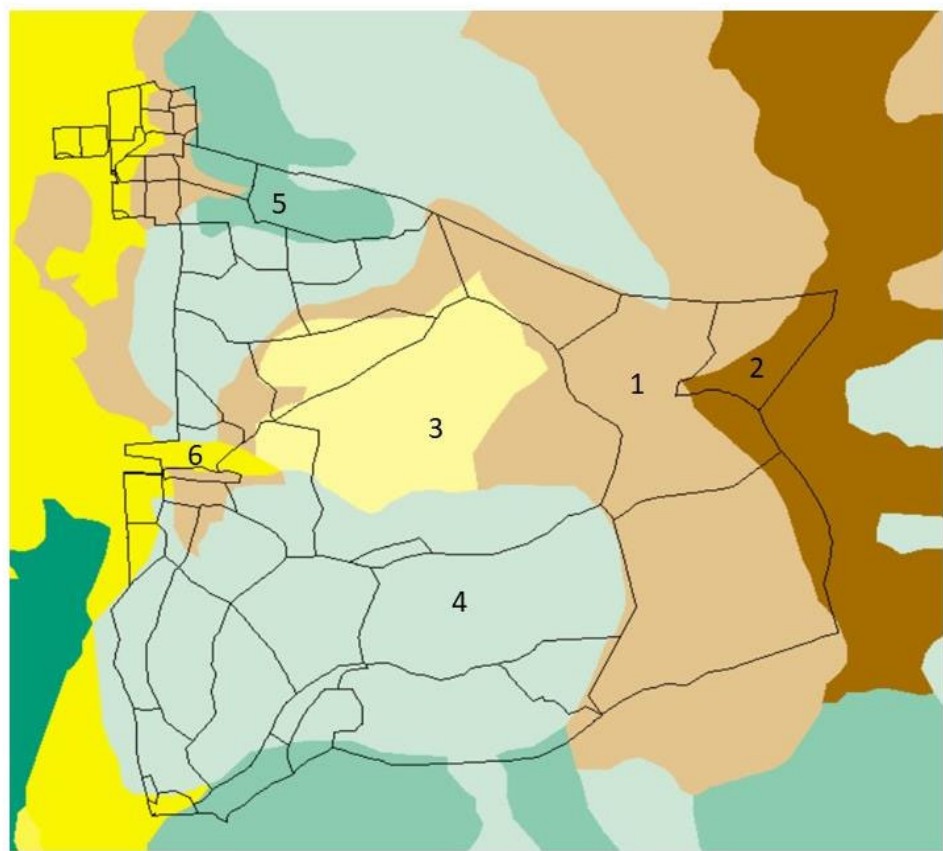

**Figure 2.** Soil classification of LUMGS derived from S-Map (Manaaki Whenua Landcare Research map at a nominal 1:50,000 scale) and the NZ National Soil Database and NZ soil classification: 1: acidic orthic Brown soils. 2: Mottled acid Brown soils. 3: Rocky Raw soils. 4: Pedal immature Pallic soils. 5: Typic argillic Pallic soils. 6: Typic fluvial Recent soils.

The grazing management of LUMGS varies from paddock to paddock and was defined 25 years ago with a productivity focus based on paddock vegetation characteristics (plant species and biomass). The number of animals (50 to 65 animals from a total of around 3000 Merino sheep) and grazing duration in each paddock (between 30 and 300 days) varies according to the paddock size, location, seasonality, and productivity stage of animals (e.g., lambing, weaning). In some paddocks, around 20 cows also graze with sheep. Most of the paddocks used are in low (<900 m) to middle (900–1100 m) altitude areas while some upper paddocks (>1100 m) have not been used in the last four years. Of a total of 48 paddocks, we restricted the model to run over 18 paddocks as they are in the high-country area of the station and are representative of the landscape heterogeneity of LUMGS. Three of the paddocks located in the high-country area belong to the New Zealand Department of Conservation, so they were not used in the model. Other paddocks are in flat areas (renewed pasture or irrigated area) or used for other purposes (winter crop, silage), rather than for grazing animals. The description of the 18 paddocks current grazing management is shown in Table 1.

**Table 1.** Details of grazing management applied to the paddocks of Lincoln University Mount Grand Station, Hawea, Central Otago, New Zealand: altitude classification [1], area (ha), stock type (ST), stock number (SN), stock density (SD) in the number of animals per hectare (au/ha), occupation period (OP) in days and season of the year used.

| Paddock | Altitude | (ha) | ST | SN | SD | OP | Season |
|---|---|---|---|---|---|---|---|
| 6 | Low | 19 | Sheep | 100 | 5.26 | 60 | Spring |
| 7 | Low | 25 | Sheep | 120 | 4.8 | 150 | Spring/Summer |
| 11 | Low | 11 | Sheep | 100 | 9.09 | 60 | Spring |
| 12 | Mid | 32 | Sheep | 500 | 15.62 | 30 | Summer |
| 13 | Low | 63 | Sheep | 250 | 3.96 | 300 | Year long |
| 13 | Low | 63 | Cattle | 20 | 0.31 | 360 | Year long |
| 14 | Low | 55 | Sheep | 200 | 3.63 | 60 | Spring |
| 14 | Low | 55 | Sheep | 250 | 4.54 | 150 | Summer/Autumn/Winter |
| 15 | Low | 75 | Sheep | 150 | 2 | 240 | Spring/Summer/Autumn |
| 15 | Low | 75 | Cattle | 20 | 0.26 | 360 | Year long |
| 16 | Low | 30 | Sheep | 150 | 5 | 270 | Spring/Summer/Autumn |
| 19 | Mid/High | 181 | Sheep | 225 | 1.25 | 150 | Summer/Autumn |
| 25 | High | 235 | None | | | | |
| 27 | Low | 55 | Sheep | 100 | 1.81 | 120 | Spring/Summer |
| 29 | Low | 42 | Sheep | 150 | 3.57 | 240 | Spring/Summer/Autumn |
| 30 | Mid | 55 | None | | | | |
| 36 | Mid/High | 167 | None | | | | |
| 40 | Low | 29 | Sheep & cattle | 120:20 | 6.89 | 60 | Spring |
| 43 | Low/Mid | 53 | Sheep | 250 | 4.71 | 270 | Spring/Summer/Autumn |
| 59 | Mid | 65 | Sheep | 300 | 4.61 | 90 | Summer |
| 60 | High | 63 | Sheep | 400 | 6.34 | 150 | Summer/Autumn/Winter |

[1] Low altitude: <900 m, mid altitude: 900–1100 m, and high altitude: >1100 m.

## 2.2. Model Description

The model of Donovan and Monaghan (2021 [14]) was used to simulate and understand the impact of different grazing strategies defined by different stock densities, occupation periods, and stock types on the spatio-temporal susceptibility of LUMGS soils to erode. The model is based on a geospatial layer of New Zealand land cover used to locate and distinguish land use (grazed or ungrazed areas) and land cover types (low producing or high producing grasslands) [29]. The New Zealand land cover is created from the combination of two national databases, the 2016 Land Use and Carbon Analysis System (LUCAS). Satellite imagery from Sentinel-2 in 2018 was also used to identify winter-forage crop paddocks across New Zealand. Most of the paddocks are classified as "low-producing grassland" for land cover, except for paddocks 13 and 25 classified as "Exotic Forest", and paddock 15—"Kanuka and/or Manuka". The New Zealand Fundamental Soil Layer (FSL) was used to derive New Zealand soil physical (superficial macroporosity, particle size classes, gravel content, drainage class, and superficial permeability), phosphate retention and organic matter content) to calculate inherent soil erodability (k). A national digital elevation model (DEM) was used as the basis for a 15 m$^2$ representation of topography, including elevation, slope steepness (angle), and flow accumulation (i.e., slope length). Lastly, climate data were derived from rainfall grids representing monthly averages over the 30 years from 1981–2010. All the information was overlaid programmatically using python. Soil properties from each paddock used as input for the model are shown on Table 2.

The damage from treading to soil physical properties was modelled as a change (in percentage) on soil permeability and structural vulnerability under different soil type, soil moisture conditions, and clay, organic matter, and P content. The input variables used to associate the soil damage with grazing intensity were stocking density, stock hoof pressure, treading duration and history, average seasonal soil moisture content, and soil clay content. Measurements of pre-and post-grazed soil attributes of microporosity, bulk density and permeability were compiled from literature to reflect the magnitude of the treading impact to soil physical attributes. Soil susceptibility for compaction and pugging during

treading was adjusted for different degrees based on clay and soil moisture. Changes in soil permeability and structural vulnerability were then integrated into the RUSLE framework as a post-treading soil erodibility factor.

**Table 2.** Soil properties from paddocks of Lincoln University Mount Grand Station derived from the New Zealand Fundamental Soil Layer (FSL) database.

| Paddock | Sandy (%) | Silt (%) | Clay (%) | Gravel (%) | Permeability | P Retention (%) | OM Content (%) |
|---------|-----------|----------|----------|------------|--------------|-----------------|----------------|
| 6 | 0.5 | 0.25 | 0.25 | 0.03 | M/S | 13 | 0.07 |
| 7 | 0.5 | 0.25 | 0.25 | 0.03 | M | 13 | 0.07 |
| 11 | 0.5 | 0.25 | 0.25 | 0.03 | M | 13 | 0.07 |
| 12 | 0.5 | 0.25 | 0.25 | 0.03 | M | 13 | 0.07 |
| 13 | 0.5 | 0.25 | 0.25 | 0.03 | M | 13 | 0.07 |
| 14 | 0.5 | 0.25 | 0.25 | 0.03 | M | 13 | 0.07 |
| 15 | 0.5 | 0.25 | 0.25 | 0.03 | M | 13 | 0.07 |
| 16 | 0.5 | 0.25 | 0.25 | 0.03 | M | 13 | 0.07 |
| 19 | 0.5 | 0.25 | 0.25 | 0.03 | M | 13 | 0.07 |
| 25 | 0.5 | 0.25 | 0.25 | 0.03 | M | 35 | 0.07 |
| 27 | 0.5 | 0.25 | 0.25 | 0.03 | M | 13 | 0.07 |
| 29 | 0.2 | 0.6 | 0.2 | 0.1 | M | 35 | 0.07 |
| 30 | 0.5 | 0.25 | 0.25 | 0.03 | M | 35 | 0.07 |
| 36 | 0.5 | 0.25 | 0.25 | 0.03 | M | 13 | 0.07 |
| 40 | 0.5 | 0.25 | 0.25 | 0.03 | M/S | 13 | 0.07 |
| 43 | 0.5 | 0.25 | 0.25 | 0.03 | M/S | 13 | 0.07 |
| 59 | 0.2 | 0.6 | 0.2 | 0.1 | M | 35 | 0.07 |
| 60 | 0.2 | 0.6 | 0.2 | 0.1 | M | 35 | 0.07 |

The grazing impact on the ground cover was modelled by adjusting the cover and management factor for different stock types as a percentage of the area to be exposed and plant residues after grazing [14]. The cover management factor considered in the RUSLE equation form varies from a value close to zero, meaning well-protected land cover, to 1, indicating barren areas [21].

Soil susceptibility for inherent surface erosion and grazed surface erosion was finally calculated as the product of all the factors previously described for each season, the total sum of which reflects annual surface erosion.

The model of Donovan and Monaghan (2021 [14]) was validated by comparing outputs with data of peer-reviewed research articles that measured soil loss from grazed lands over spatial and temporal scales across New Zealand.

*2.3. Grazing Management Simulations*

High country stations are more prone to superficial erosion due to topographic factors and soil is a key component of grassland health. Thus, grazing management applied to those environments should prioritise the soil and adapt the number of animals and period of grazing to the capability of the land to support the treading that will occur. On New Zealand hill country farms, grazing management for sheep production is usually continuous for two months, or over the season, with a stock density (number of animals that are kept on a given unit of area, SD) varying between 6.2 and 16 au/ha according to pasture quality and availability [30–33]. Depending on soil type and season, some paddocks are still ungrazed or grazed at low intensities to reduce treading damage [34,35].

As the impact of grazing is highly contextual and the effect of a particular grazing management on one farm may be different elsewhere, we took the modelling approach to better understand the spatial-temporal response of LUMGS soils to grazing and treading. We first ran scenarios representing ungrazed pastures for each paddock and each season of the year to assess the natural annual soil loss by erosion of the station, and a scenario representing the current standard management taken in each paddock (status quo). Then,

we applied different grazing strategies (grazing intensity and livestock types) to estimate each paddock resistance and threshold to treading.

Results from ungrazed pasture scenarios were plotted against the aspect, slope angle, and altitude factors of LUMGS to illustrate the landscape heterogeneity of the station and its influence on soil erosion. For this purpose, each factor was classified into three categories. The aspect was classified as sunny aspect—north, northeast, and northwest face; moderate sunny aspect—west and east faces; and shady aspect—south, southeast, and southwest hillside faces. The slope angle was divided into 0–15°, 15–30°, and higher than 30°. Lastly, altitude was classified as high altitude (>1100 m), mid-altitude (900–1100 m), and low altitude (<900 m).

To assess the response of each paddock to grazing intensity, we modelled the effect of different SD and grazing occupation periods (OP), based on SD and OP commonly used on New Zealand hill country farms. Thus, we took the median value of SD found in the literature (12 ewes/ha) and varied this SD in 50% steps to get to a range from 3 to 27 ewes/ha for SD (3, 6, 12, 18, 27 ewes/ha). For OP, we again varied in 50% steps from the usual period according to the literature (60 days) to get to a range from 1 to 303 days, and included a whole year—365 days, for final scenarios of 1, 3, 7, 15, 30, 60, 90, 135, 202, 303, and 365 days.

As some paddocks on hill country farms are ungrazed or grazed at low intensities during wet seasons to reduce treading damage, and most of the LUMGS soils are Pallic, which are very sensitive to treading, during wet seasons [36], we assessed the temporal variation of paddocks response to each scenario. To do so, we adjusted the number of days in each OP for a maximum of 90 days, the average considered for each season, and eliminated the scenarios in which OP was less than 1 day, which resulted in OP of 1, 3, 7, 15, 22, 33, 50, 75, and 90 days, and a total of 45 scenarios (5 SD and 9 OP) (Table 3) applied for each season (45 times, 4 seasons—180 scenarios). To explore the impact of those grazing intensity strategies with different stock types (ST), all the 180 scenarios were applied for sheep, beef, and deer, which yielded a total of 540 scenarios, plus the ungrazed and the status quo scenarios, producing a total of 545 scenarios.

**Table 3.** Description of the scenarios (different occupation periods (OP) in days and stock densities (SD) as au/ha) modelled over Lincoln University Mount Grand Station.

| Scenario | OP (days) | SD (au/ha) |
|---|---|---|
| 1 | 1 | 3 |
| 2 | 1 | 6 |
| 3 | 1 | 12 |
| 4 | 1 | 18 |
| 5 | 1 | 27 |
| 6 | 3 | 3 |
| 7 | 3 | 6 |
| 8 | 3 | 12 |
| 9 | 3 | 18 |
| 10 | 3 | 27 |
| 11 | 7 | 3 |
| 12 | 7 | 6 |
| 13 | 7 | 12 |
| 14 | 7 | 18 |
| 15 | 7 | 27 |
| 16 | 15 | 3 |
| 17 | 15 | 6 |
| 18 | 15 | 12 |
| 19 | 15 | 18 |
| 20 | 15 | 27 |
| 21 | 22 | 3 |
| 22 | 22 | 6 |

**Table 3.** *Cont.*

| Scenario | OP (days) | SD (au/ha) |
|---|---|---|
| 23 | 22 | 12 |
| 24 | 22 | 18 |
| 25 | 22 | 27 |
| 26 | 33 | 3 |
| 27 | 33 | 6 |
| 28 | 33 | 12 |
| 29 | 33 | 18 |
| 30 | 33 | 27 |
| 31 | 50 | 3 |
| 32 | 50 | 6 |
| 33 | 50 | 12 |
| 34 | 50 | 18 |
| 35 | 50 | 27 |
| 36 | 75 | 3 |
| 37 | 75 | 6 |
| 38 | 75 | 12 |
| 39 | 75 | 18 |
| 40 | 75 | 27 |
| 41 | 90 | 3 |
| 42 | 90 | 6 |
| 43 | 90 | 12 |
| 44 | 90 | 18 |
| 45 | 90 | 27 |

The model outputs provide the yield of soil loss (t/ha/year) and the total of soil loss in each paddock per year, which is the accumulation of soil loss from every season. Soil loss was plotted against grazing intensity at a paddock level for the visualisation of the trend between the factors and filtering of scenarios. The inflection point of the curve was calculated by using the inflection package on R software version 3.4.0 [37]. The inflection point represents the point of significant changes, which in this case would indicate maximum values of grazing intensity (threshold) that paddocks could support for avoiding increased soil loss. We used the inflection point to select the best scenarios for each paddock based not only on productivity but on their susceptibility to superficial erosion. The use of the best scenario for each paddock would be a strategy for holistic grazing management for increased grassland health under pastoral livestock production systems.

### 3. Results

Figure 3 presents the soil annual loss by superficial erosion due to the inherent landscape condition of LUMGS paddocks. The difference in colours indicates a high variance across paddocks in terms of soil loss due to landscape heterogeneity. This variance was less different between seasons. The variation from the less susceptible paddock to the most susceptible paddock to soil erosion was 0.02 to 0.69 t/ha/year in autumn, 0.01 to 0.55 t/ha/year in spring, 0.02 to 0.57 t/ha/year in summer, and 0.01 to 0.73 t/ha/year during the winter.

The relationship between annual soil loss by natural erosion from ungrazed pasture scenarios (t/ha/year) and aspect, slope, and altitude are shown in Figures 4–6. Soil loss was greater for sunny aspects, areas with 15–30° of slope angle, and areas located at high altitudes.

The status quo scenario resulted in a total annual soil loss of 6.65 t/ha/year, but the rate of soil loss of individual paddocks was divergent (Figure 7). Due to a variance in landscape condition and grazing management (different SD and OP as shown on Table 1), soil loss varied from 0.119 to 1.793 t/ha/year.

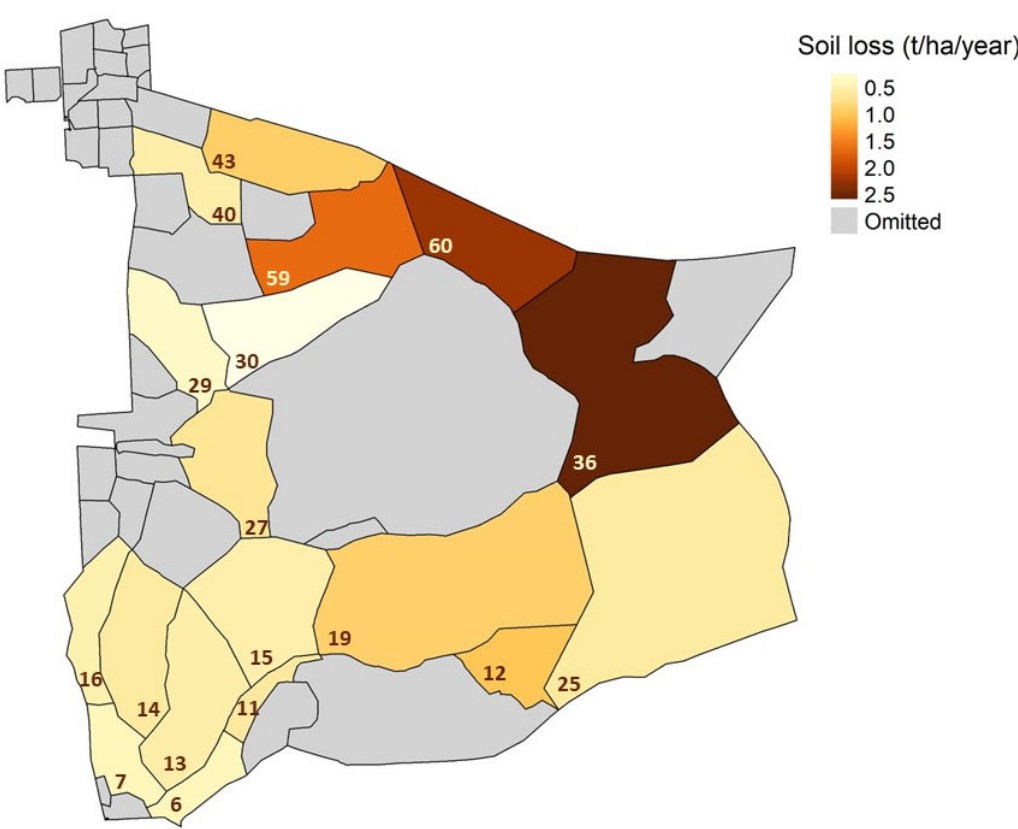

**Figure 3.** Model output of annual soil loss (t/ha/year) by erosion due to the inherent landscape condition of Lincoln University Mount Grand Station paddocks (ungrazed pasture scenarios).The numbers are the identification of the paddocks. Omitted data represents paddocks that were not modelled.

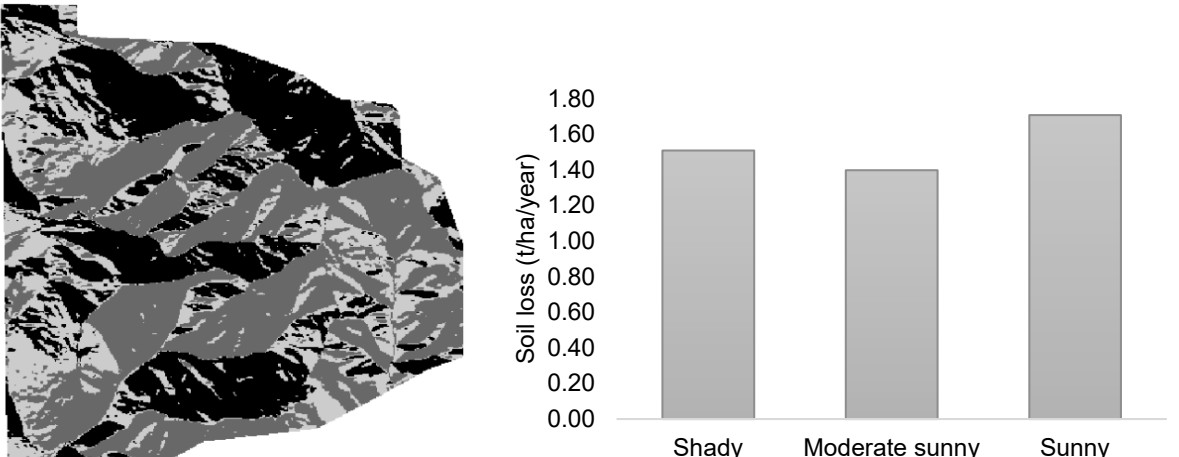

**Figure 4.** Relationship between annual soil loss by natural erosion from ungrazed pasture scenarios (t/ha/year) and aspect classification of Lincoln University Mount Grand Station. Sunny aspect—north, northeast, and northwest face; moderate sunny aspect—west and east faces; and shady aspect—south, southeast, and southwest hillside faces. The map to the left illustrates the zonally averaged values of soil loss by erosion in a gradient colour according to the three considered aspect categories over the landscape. The dark colour means higher soil loss by erosion (sunny aspect) and lighter colour means lower soil loss by erosion (moderate sunny aspect).

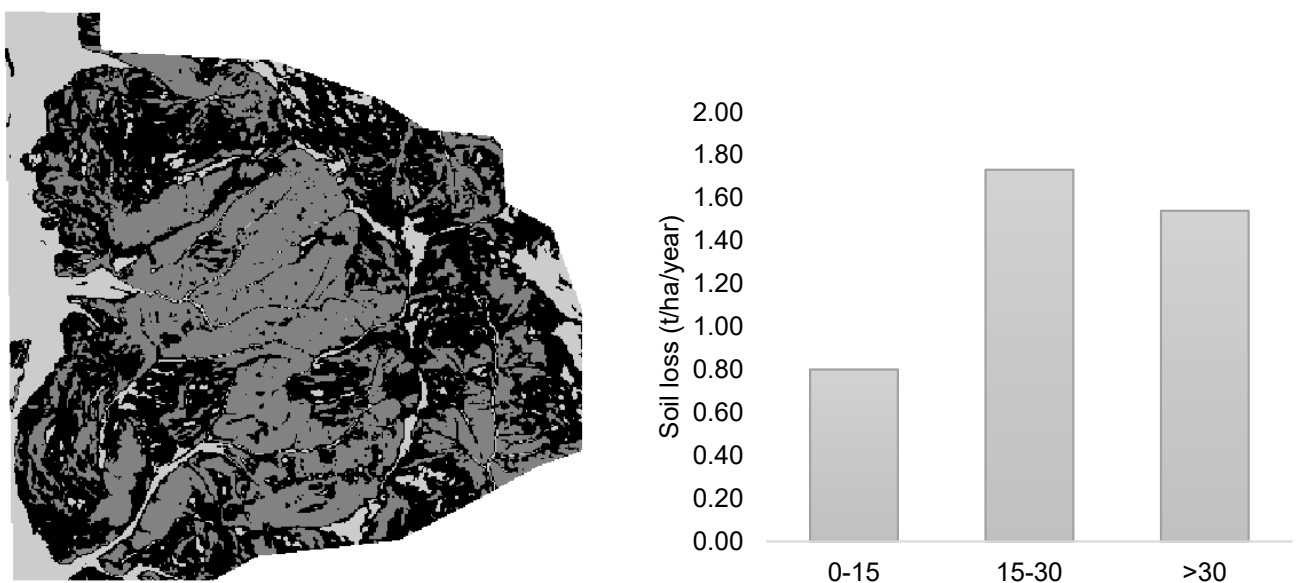

**Figure 5.** Relationship between annual soil loss by natural erosion from ungrazed pasture scenarios (t/ha/year) and slope angle (0–15°, 15–30°, and higher than 30°) of Lincoln University Mount Grand Station. The zonally averaged values of soil loss by erosion are in a gradient colour according to the three considered slope categories over the landscape. The dark colour means higher soil loss by erosion (15–30°) and lighter colour means lower soil loss by erosion (0–15°).

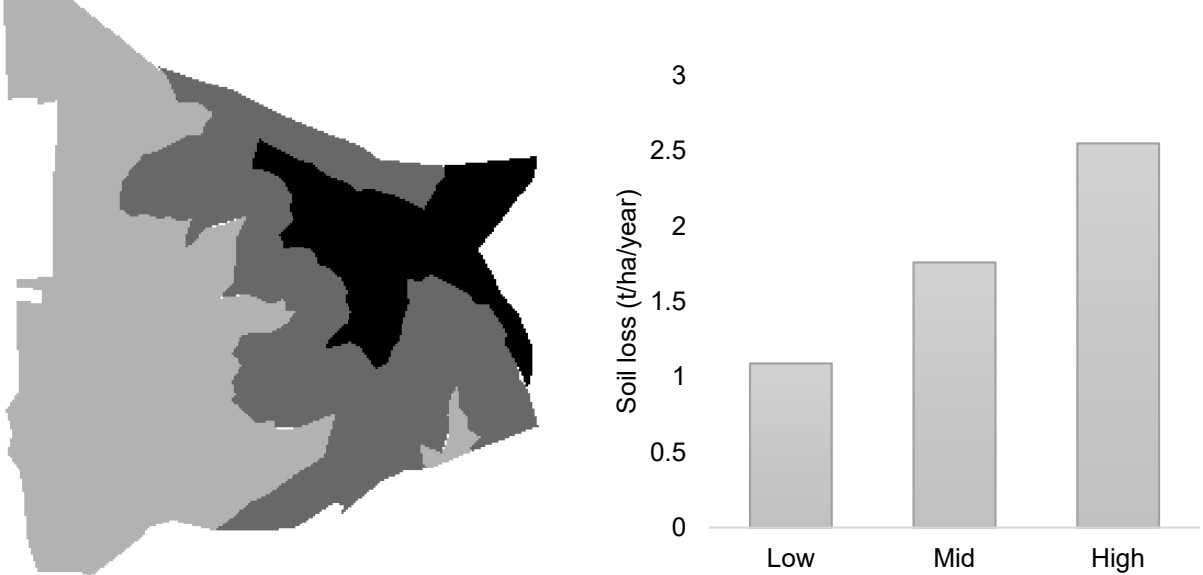

**Figure 6.** Relationship between annual soil loss by natural erosion from ungrazed pasture scenarios (t/ha/year) and altitude classification of Lincoln University Mount Grand Station; high altitude (>1100 m), mid-altitude (900–1100 m), and low altitude (<900 m). The zonally averaged values of soil loss by erosion in a gradient colour according to the three considered altitude categories over the landscape. The dark colour means higher soil loss by erosion (high altitude) and the lighter colour means lower soil loss by erosion (low altitude).

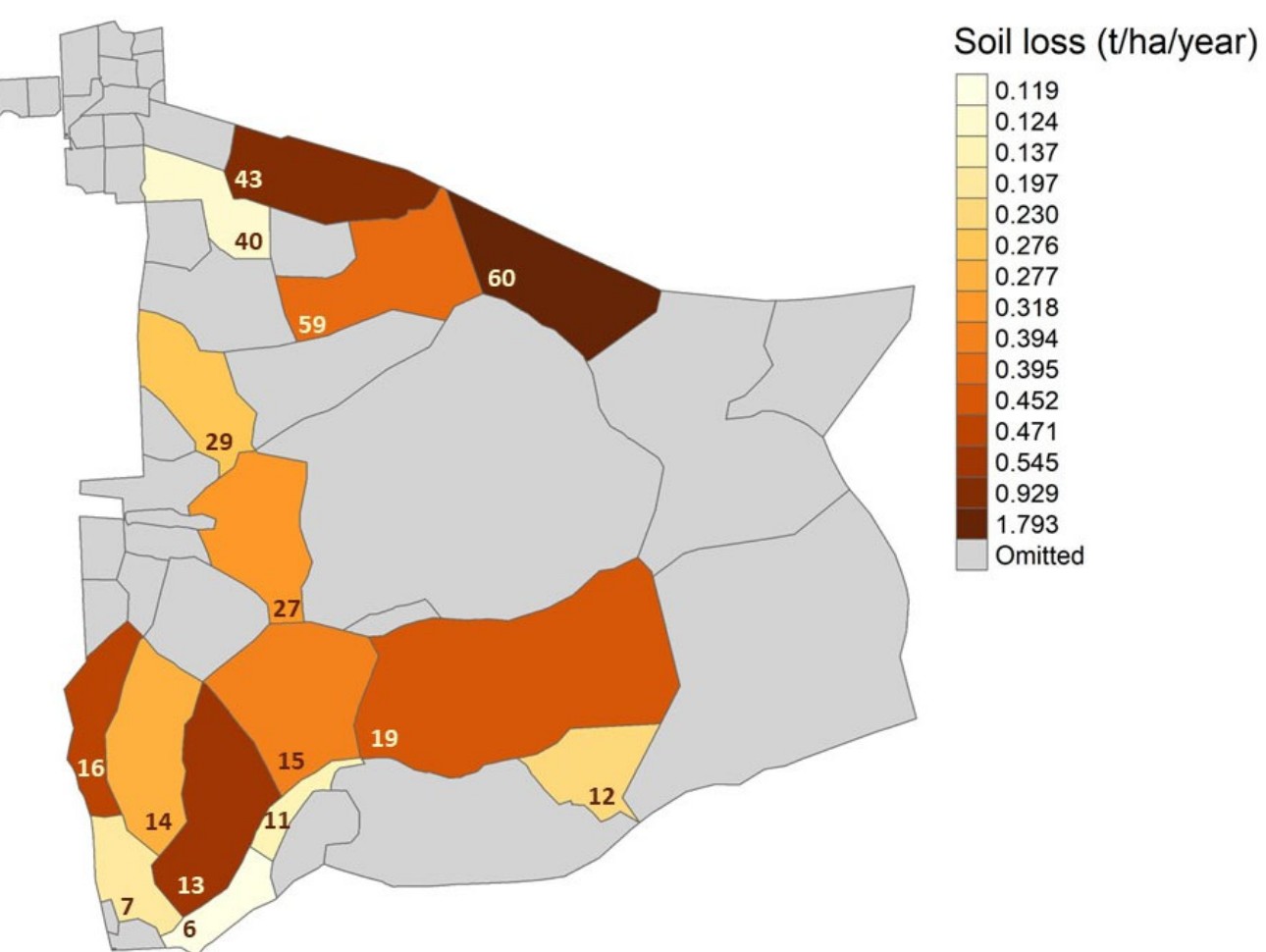

**Figure 7.** Model output of annual soil loss (t/ha/year) of Lincoln University Mount Grand Station paddocks by erosion due to grazing activity representing the total use of each paddock throughout the year according to the current grazing management of the station. Dark colours mean higher soil loss. The numbers are the identification of the paddocks. Omitted data represents paddocks that were not modelled.

When plotting soil loss against SD and OP for different ST (Figure 8), we can see that soil loss increases as both SD and OP increase and is, in general, higher for cattle. However, the slope of the trend is more noticeable for OP, indicating that increases in OP are more sensitive and more detrimental to soil loss than SD. This relationship between soil loss changes and SD and OP was conducted by considering average values from all scenarios (except standard management and ungrazed pastures scenarios), paddocks, and seasons.

The seasonal variability of soil superficial erosion in response to grazing and treading differed across paddocks. Thus, we created a map (Figure 9) indicating the best season for paddocks to be used according to their seasonal susceptibility to erosion. Figure 9 also shows the possible average rate of annual soil loss (t/ha/year) that would occur for minimising soil loss by erosion if paddocks were used in the recommended season. These values were calculated considering the average of soil loss from all scenarios and ST ran in the model and only consider the yield of soil loss caused by grazing and treading. Most paddocks were less susceptible to erosion during spring, followed by summer. Only two paddocks showed lower susceptibility during winter compared to other seasons, and no paddock showed lower susceptibility during autumn.

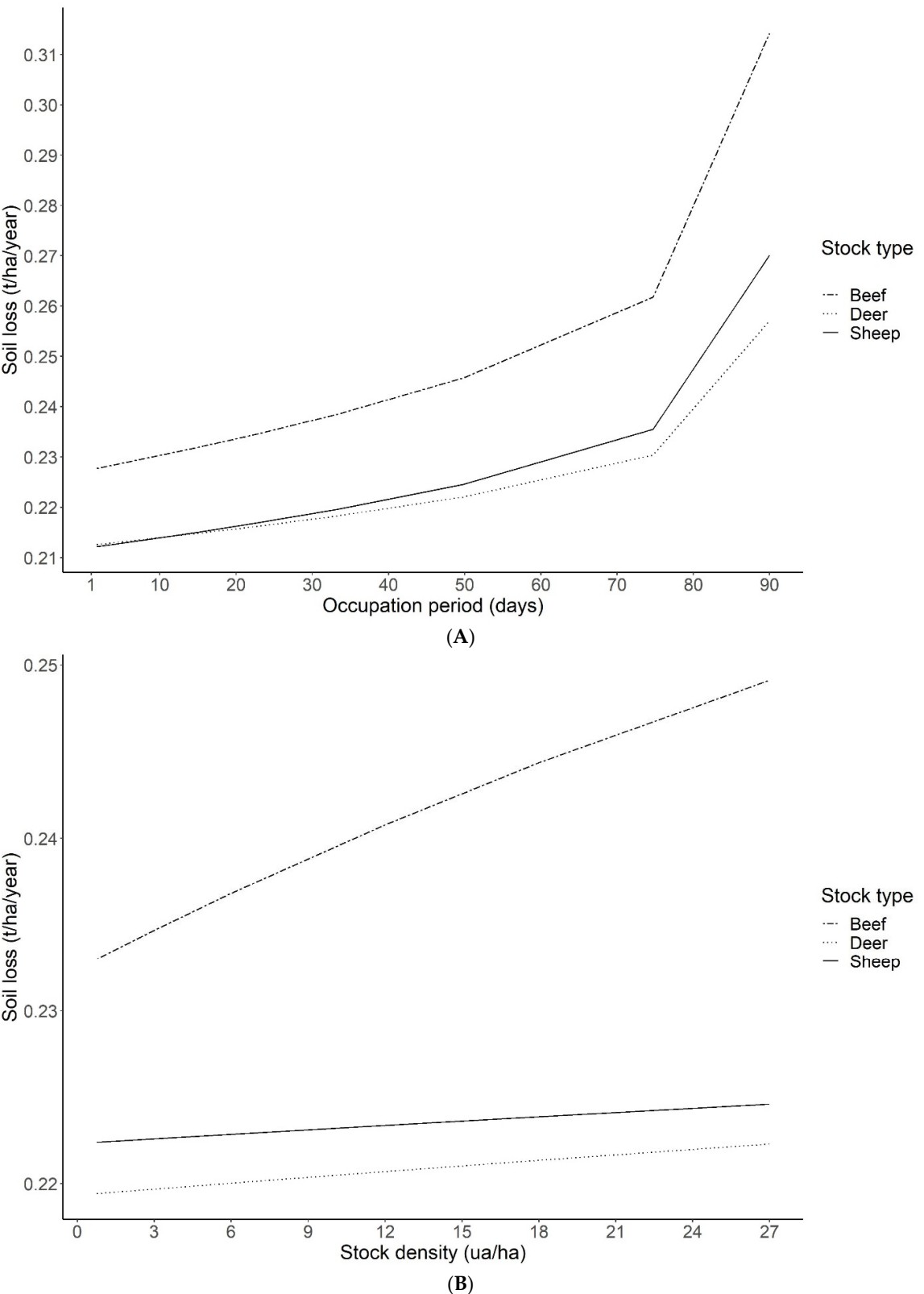

**Figure 8.** Soil loss erosion (t/ha/year) according to increase in occupation period (days) (**A**) and stock density (au/ha) (**B**) for different stock types across all Lincoln University Mount Grand Station paddocks, scenarios, and seasons considered in the model.

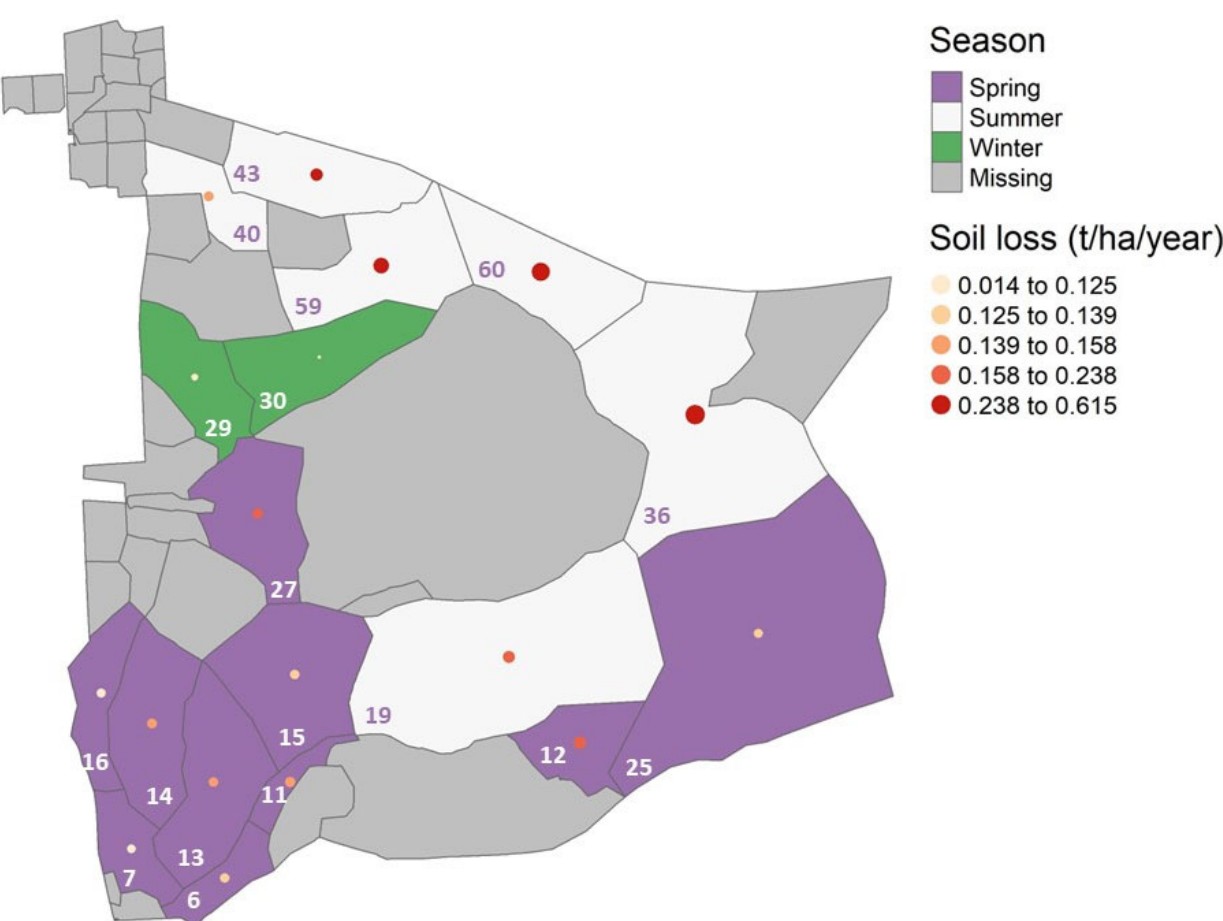

**Figure 9.** Indication of the best season that paddocks of Lincoln University Mount Grand Station should be used for grazing and the average annual soil loss (t/ha/year) it would occur considering all scenarios and stock type ran in the model for minimizing soil loss by erosion due to grazing impact. The use of paddocks in autumn is not indicated as no paddock showed lower susceptibility for this season. The numbers are the identification of the paddocks. Omitted data represents paddocks that were not used in the model.

The inflection point was used as a criterion to find the maximum limit of increased grazing intensity that paddocks can support while soil erosion is controlled. Accordingly, we found that OP should be no longer than 50 days and SD should be no greater than 27 au/ha. There are several options of grazing intensity that can be applied to each paddock based on different combinations of OP and SD, but we selected the best alternatives at the inflection point that maximise the number of animals while minimise soil loss for each paddock, season, and ST (Table S1).

## 4. Discussion

### 4.1. Soil Erosion from Ungrazed Scenarios

A modelling approach was taken to better understand the spatial-temporal variance of LUMGS soils susceptibility to erosion. This was conducted by manipulating variables that define grazing intensity (OP and SD) and strategy (ST) to determine each paddock resistance and threshold to treading as a tool to design grazing management that improves grassland health by reducing soil erosion. Natural soil erosion in LUMGS is highly variable spatially (Figure 3) and slightly variable temporally. This is explained by the heterogeneity of the landscape with varied inherent and dynamic attributes that characterize a wide distribution of soil resilience and susceptibility to erosion [38]. Aspect, slope angle, and

elevation (Figures 4–6) are likely to be the main factors influencing LUMGS paddock heterogeneity in terms of soil characteristics.

We can see that greater natural soil erosion generally occurs in LUMGS paddocks which are north (sunny) facing at mid to high elevations. The increased solar radiation and higher evaporation in sunny-facing aspects induce microclimatic conditions in soil, such as lower soil moisture content, that influences soil nutrients, organic matter, microbial activity, and consequently, the above-ground biomass [39,40]. Effects of heating are even more pronounced on steeper terrain—as in the area used in this study [39]. Soil sampling in the same study area, Pereira et al. (2021 [28]) observed lower soil moisture content and higher pH in sunny aspects compared to moderately sunny and in shadow aspects, in addition to other trends and differences in soil properties. Soil moisture, organic matter content, and vegetation cover have a positive influence on protecting soils against erosion [41,42]. Thus, areas with lower ground cover will be more susceptible to erosion [14] as observed in this study.

The areas with greater natural erosion have a greater cover management factor according to the RUSLE classification. Ground cover affects the soil erodability factor and depends on the climate and topographic factors, such as rainfall and slope, besides soil conditions, texture, organic matter, structure, and permeability [43]. Thus, ground cover explains the LUMGS spatial-temporal variability of soil erosion. Paddocks with higher natural soil loss are in the low-producing grassland areas, in which cover management factor is greater than in other areas from the farm, such as high-producing grassland or shrubland and woody areas, especially after grazing. As these paddocks are also located at higher altitudes, the slope has a great influence on the soil loss, as the greater the slope length, the greater the soil loss per unit area [21,23].

The spatial variability of LUMGS soil erosion can also be justified by inherent properties that determine the erodability of a soil. Some of the properties are soil texture, organic matter content, soil structure, and permeability, which can cause some soils to erode more quickly even if all the other influencing factors are the same [44]. The soil at LUMGS is characterized by a silt loam texture. As silt content in the soil increases, the higher its erodability, but this increase is dependent on other chemical or physical properties such as organic matter content, pH, and aggregation stability [45]. Those factors, along with the topographic factors such as aspect and slope that affect soil surface characteristics [41] explain the differences among LUMGS paddocks in terms of susceptibility to erosion [21].

*4.2. Temporal Variation of LUMGS Soil Erosion*

Autumn and winter were, in general, the seasons when more soil erosion occurred. The natural variation in erosion between seasons is due to changes in the dominance and interaction of erosive forces—wind, water, and frost [46], the proportion of vegetation cover, and the combination of all those factors with topographic parameters [23]. Rainfall is scarce and well distributed during the year in the region where LUMGS is located; however, a slight increase is noticed during autumn and winter. The wind is stronger during summer and spring, and winter is characterized by snow at the highest elevations [47]. Although not all the erosive factors are considered in the model used in this study, they may explain the temporal and spatial variation of LUMGS, thus contributing to the characterization of its soils and vegetation cover abundance. These spatial and temporal variations cause changes in soil morphologies and water content with further consequences for erosion rates [48].

Rainfall distribution and vegetation cover are accounted for in the RULSE equation, and although rainfall has been shown to be more significant than vegetation-related factors in some studies [22], others show the ground cover to predominate the soil loss signal. These factors are associated with more covered areas having greater efficacy in reducing the impact of rainfall and thus, runoff [14]. Nevertheless, the weight of the factors influencing soil erosion may be different in different seasons. For instance, Ochoa et al. (2016 [43]) found during the wet season, ground cover and rainfall factors may exert more influence on soil erosion than topographic and soil erodability factors, while the opposite would

occur during the dry season. During summer, infiltration rates of rainfall are favoured due to the cracks and macropores formed due to dry conditions [49], while the wetter conditions in autumn and winter cause the infiltration rates to reduce [48]. The lower soil moisture content during summer also favours deep infiltration of rain, which positively affects soil runoff [50]. During winter, the soil surface is covered by snow and bare areas may be frozen, which keeps particles together and protects the soil against wind erosion. However, water erosion due to the snowmelt runoff may be of greater influence, especially in upslope areas [46]. This event in addition to the greatest soil water content during winter may explain the high erosion rates from this season. In spring, the wind is a more dominant erosive force, as the loose and weak topsoil becomes a material source for erosion after the snow melt period [46].

Both the combination of rainfall distribution and vegetation cover and the effects of erosive forces are affected by topography. Wind erosion is more pronounced in increasing slope gradients, due to the reduced vegetation cover, especially in convex locations of the slope [51], while water erosion is more severe in lower parts of the hillslope than in upper parts [46], but reduced vegetation cover in steeper slopes causes bare areas to be more susceptible to water erosion due to the lower infiltration rates [50]. All the aforementioned factors help to explain the seasonal variability in the erosion of LUMGS soils, with a stronger impact from rainfall and vegetation cover and their effect on soil properties, as both are considered in the model used in this study. Furthermore, understanding the complex relationship between temporal and spatial variations of LUMGS (topography, erosive factors) contributes to the annual distribution for the use of LUMGS paddocks.

### 4.3. Grazing Management Strategies

Besides soil and topographic characteristics, grazing management contributes to the variation in soil loss by superficial erosion in LUMGS, which despite being divergent across paddocks, is mainly characterized by animals grazing at low SD and long OP (Table 1). Notwithstanding, the model outcomes suggest that, overall, reducing OP seems to be a strategy to mitigate soil erosion. Although soil erosion increased with grazing intensity (Figure 8), we noticed an exponential tendency relation between soil loss and OP especially after 75 days, while a more constant relationship was found for soil loss and SD. This suggests that increasing OP results in greater soil erosion, which may be related to the decrease in ground cover over time that reduces the soil protection against erosion, resulting in lower infiltration rates and lower organic matter input. Controlling the duration of grazing allows plants to recover during a rest period after defoliation by animals, which increases and maintains ground and foliar cover and infiltration rates. This strategy is more efficient in reducing runoff compared to continuous grazing. This is consistent with a study conducted in Australia, in which runoff reduced over time even in paddocks located on a steep hill slope, while an increasing trend was noticed in the continuous system [52].

Increased SD does not seem to be as detrimental as our inflection point analysis suggested SD up to 27 au/ha. However, increased SD with long OP can negatively affect soil health. Pulido et al. (2018 [6]) reported that an increased rate of SD in a continuous grazing system resulted in a greater proportion of bare ground due to excessive defoliation, with additional increases in soil bulk density that is a barrier for pasture production. In contrast, high SD in a short-duration rotational grazing system was beneficial for pasture productivity and soil health, as noticed by a greater stock of C, N, and OM than in a continuous system [53]. Similar results were found by Sanjari et al. (2008 [54]) where time-controlled grazing followed by long rest periods resulted in greater levels of C and N in soil compared to continuous grazing, and increased above-ground organic matter, that protects the soil from treading. In our study, the selection of the best-compromise scenarios with the inflection point indicates a maximum OP of 50 days, as, after this point, the increase in soil erosion increases significantly. Therefore, we can infer that for LUMGS

an increased SD with long OP is not ideal, but increased SD at a threshold OP of 50 days seems to be plausible for soil health.

Soil erosion at LUMGS was greater for cattle, which is expected as those animals are heavier and exert greater downward pressure on the soil per unit area from hoof contact with consequently greater treading effect than sheep and deer [7,55]. Greater impacts on the physical properties of soil by cattle were also noticed by Cournane et al. (2011 [56]) and Houlbrooke et al. (2006 [57]). However, in Cournane et al. (2011 [56]), differences in soil surface were detected by season rather than ST, indicating that the time when paddocks are grazed is the greatest factor of influence. Cournane et al. (2011 [56]) also indicated that grazing practices such as rotational grazing, restricted grazing or decreased SD adjusted for the season are more efficient in reducing soil loss than choosing the ST per se. We can see similar outcomes in our results. Although cattle are the most detrimental ST, same SDs and OPs as for sheep and deer were selected for cattle as the best scenarios (but cattle would require different combinations of both factors, which would imply in lower grazing intensity compared to sheep and deer). Accordingly, sheep grazing paddocks during seasons in that soil erosion is greater could result in similar soil loss as paddocks grazed by cattle in the most convenient seasons for reducing soil erosion. Therefore, we suggest as a grazing strategy that sheep and deer, as the less detrimental ST to soil erosion, should graze the paddocks that are more naturally susceptible to erosion while cattle should graze the paddocks that are more naturally resistant to soil erosion at SD and OP properly adjusted.

*4.4. Final Considerations*

The model outcomes display the spatiotemporal variance of LUMGS paddocks response to soil erosion and grazing intensity, thus, suggesting that grazing management should be adjusted at the paddock and season level rather than a generalized approach of managing animals to increase productivity. The scenarios selected by using the inflection point demonstrate that grazing intensity in each paddock could be potentially increased without further consequences for soil erosion. Overall, the status quo keeps animals grazing continuously in the same paddock for months at low SD. Based on our results, we suggest as alternative grazing management to the status quo and at similar levels of pasture utilization, a faster rotativity of animals with reduced resident time in each paddock (15, 20, 30, or 50 days) and greater grazing pressure (medium to high SD rates: 6, 12, 18 or 27 au/ha). Moreover, paddocks should be used during the seasons that are more convenient for reducing soil erosion (according to each paddock seasonal susceptibility to erosion) and with the most suitable ST. For instance, as beef cattle are the most detrimental ST in the context of soil erosion, followed by deer, and autumn is the season when more soil erosion occurs, the most resistant paddocks to erosion should be grazed in autumn, and from them, the most resistant ones should be grazed by cattle. Our results suggest that grassland health can be enhanced in LUMGS by applying flexible and adaptable grazing management that implies in the use of context-adjusted grazing strategies to increase the productivity of the farm while controlling soil erosion from treading.

While the grazing management simulations were not validated, the model used in this study is based on New Zealand database and validated by comparing outputs with data of peer-reviewed research articles that measured soil loss from grazed lands across New Zealand across a range of spatial and temporal scales. We used the modelling approach to explore a vast number of different scenarios over LUMGS and understand their impacts on soil erosion in a spatial-temporal context, what would be impracticable on field. The modelling allows for the preliminary test of scenarios for the selection of the best-compromised ones before testing them in the field. However, we strongly recommend as a further work the application of the best-compromised grazing simulations defined here over LUMGS across a year for increasing the precision of the model.

## 5. Conclusions

We applied a geospatial livestock treading and grazing model to understand the spatio-temporal capacity of a high-country pastoral livestock production system in New Zealand to withstand grazing and treading while avoiding soil superficial erosion. The grazing model outputs indicated that LUMGS soil susceptibility to erosion is spatially and temporally variable, and extremely context dependent upon soil conditions. This variance implies that paddocks in our study area have different capacities to support grazing which also changes during seasons. Overall, soil erosion increased with grazing intensity, but increases in occupation periods appear more detrimental to the soil than increases in stock density, as noticed by an exponential tendency relation between soil loss and occupation period. Accordingly, grazing management should be based on occupation periods no longer than 50 days and medium to high stock density rates. Of all the stock types, cattle in the context of soil erosion are the most detrimental animal compared to sheep and deer. Based on the model outcomes, to preserve and enhance the health of LUMGS grasslands, grazing management should be planned by applying the appropriate stock density, occupation period, and stock type to each paddock, and grazing them at the season when their capacity to support treading is greater. We acknowledge that landscape inherent heterogeneity condition and temporal variation may be more influential for soil erosion than the grazing factors (SD, OP, and ST) per se, but high-country stations in New Zealand would benefit from a more flexible and adaptable grazing management for enhanced grassland health.

**Supplementary Materials:** The following supporting information can be downloaded at: https://www.mdpi.com/article/10.3390/soilsystems7020030/s1, Table S1: Selection of best scenarios (defined by occupation period in days, and stock density, au/ha) of Lincoln University Mount Grand Station paddocks for each season and each stock type (sheep, deer, or cattle) to maximise grazing intensity and reduce soil loss (t/ha/year), according to outputs of the Donovan and Monaghan (2021 [14]) grazing and treading model.

**Author Contributions:** F.C.P. contribute to the concept of this work in consultation with all co-authors. F.C.P. and M.D. performed the model simulations. F.C.P. analysed the data and wrote the manuscript, and M.D., C.M.S.S., S.C., T.M.R.M. and P.G. contributed to the review and edition. All authors have read and agreed to the published version of the manuscript.

**Funding:** This research was funded by the Centre of Excellence, Designing Future Productive Landscapes, Lincoln University (4350 AGLS—INT 4908 AGLS—113043), and by the John Barnes Postgraduate Scholarship from Lincoln University.

**Data Availability Statement:** The data presented in this study are available in the supplementary material.

**Conflicts of Interest:** The authors declare no conflict of interest.

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
