# Peer review of "A Geospatial Modelling Approach to Understand the Spatio-Temporal Impacts of Grazing on Soil Susceptibility to Erosion"

_soilsystems, doi:10.3390/soilsystems7020030_

Round 1

Reviewer 1 Report

The manuscript addresses a topic that will be of interest to readers. It demonstrates the influence of spatial and temporal changes in grazing related to soil attributes. It is an interesting paper to read since it applies interesting scenarios. The manuscript is a viable candidate for publication; however, it was difficult to follow. I recommend major revisions. The way to present the figures is not clear enough, and there are many flasks in material and methods.

Material and method

Lines 104-105 give more detail about Donovan and Monaghan’s model.

It is recommendable to start the material and method section with the study area rather than explaining the models.

Which software did you use to overlap the information GIS?

124 -125 Please clarify these lines.

151 rewrite this sentence

191 SD was defined two times. In lines 191 and  219.

231 The axis titles and legends are confusing. It would be helpful to be consistent throughout the text using t/ha/year as a rate of soil loss and soil loss ( t/ha). Be consistent in the text and figures.

241 surficial or superficial?

239 -242 Please clarify this paragraph. I believe it is out of context in this Material and Methods.

Was the RUSLE Model validated by comparing the erosion measurements at the field?

Results

I missed a table including the soil properties according to the studied paddock. pH, soil C, moisture, etc

Please modify SOIL LOSS (T/HA) in the axes Y as Soil Loss (t/ha). Avoid using Uppercase. I got confused because you define Soil Loss in (t/ha) and then (t/ha/year) as a Soil Loss too. Please check this point.

I think some figures could be moved to supplementary materials.

I don’t understand the contribution of Figure 3B.

Discussion

Are important parts of the argument poorly supported that are hard to relate to the figures

Reviewer 2 Report

The present study seems to seems to be meaningful. However, the introduction on model validation is lacking, which reduces the reliability of the model results. And writing needs further improvement.

1.     title should be corrected

2.     line22: Only grazing intensity?

3.     Introduction: why use this model? please explain it in details.

4.     Add a paragraph to describe the study area;

5.     Model introductions should focus on the situations relevant to this study;

6.     Add a research roadmap

7.     Results: add validation of simulation results

8.     Discussion: Divide the discussion into different modules, for example, 4.1 Uncertainty,4.2 Grazing effect, 4.3 Comparison with other studies

9.     Conclusions: rewrite

Author Response

The authors appreciate and value the comments and suggestions of the reviewer, acknowledging that they add clarity to the presented work and enrich the message conveyed in it. We have added more details about the model used to the introduction section and improved the writing.

We have attached a document with point-by-point response to all comments. Please see the attachment.

We also submitted a new version of the manuscript. One with tracked changes and one clean version.

Round 2

Reviewer 1 Report

The manuscript has the potential to make a substantial contribution to the field. The manuscript has improved significantly after revision, but still has too many weaknesses. I still suggest some major revisions.

The abstract mentions 271 scenarios, but the procedures do not clearly describe how the authors establish those scenarios.

Adding tables including soil texture or other soil attributes as well as the scenarios, would help to follow the text easily. For example, to prepare a table including the variables you describe in lines 221 to 224. It is not clear which variables were used for each scenario

The manuscript is difficult to follow and has poorly presented data.  for example SOIL LOSS (T/HA) is sometimes in capital letters, and others "Soil Loss (t/ha)" in lowercase. In addition, data need to be presented more efficiently and no more than eight figures showing almost the same.

Author Response

The authors appreciate and value the comments and suggestions of the reviewer, acknowledging that they add clarity to the presented work and enrich the message conveyed in it. We have added more details to the material and methods section and rewritten the results and conclusion as suggested. We also had the manuscript reviewed by a professional from Lincoln university for English review. Please see attached point-to-point responses to comments.

Reviewer 2 Report

The paper is improved greatly. I think it  meets the publication requrements.

Author Response

The authors appreciate and value the comments and suggestions of the reviewer, acknowledging that they add clarity to the presented work and enrich the message conveyed in it. We have added more details to the material and methods section and rewritten the results and conclusion as suggested. We also had the manuscript reviewed by a professional from Lincoln university for English review. We are submitting a new version of the manuscript.